# STAT3 Signalling Drives LDH Up-Regulation and Adiponectin Down-Regulation in Cachectic Adipocytes

**DOI:** 10.3390/ijms242216343

**Published:** 2023-11-15

**Authors:** Michele Mannelli, Bianca Bartoloni, Giulia Cantini, Elena Nencioni, Francesca Magherini, Michaela Luconi, Alessandra Modesti, Tania Gamberi, Tania Fiaschi

**Affiliations:** Dipartimento di Scienze Biomediche, Sperimentali e Cliniche “M. Serio”, Università degli studi di Firenze, Viale Morgagni 50, 50134 Firenze, Italy; michele.mannelli@unifi.it (M.M.); giulia.cantini@unifi.it (G.C.); francesca.magherini@unifi.it (F.M.); michaela.luconi@unifi.it (M.L.); alessandra.modesti@unifi.it (A.M.); tania.gamberi@unifi.it (T.G.)

**Keywords:** adipocyte, cancer cachexia, adiponectin, STAT3

## Abstract

Cachexia is a devastating pathology that worsens the quality of life and antineoplastic treatment outcomes of oncologic patients. Herein, we report that the secretome from murine colon carcinoma CT26 induces cachectic features in both murine and human adipocytes that are associated with metabolic alterations such as enhanced lactate production and decreased oxygen consumption. The use of oxamate, which inhibits lactate dehydrogenase activity, hinders the effects induced by CT26 secretome. Interestingly, the CT26 secretome elicits an increased level of lactate dehydrogenase and decreased expression of adiponectin. These modifications are driven by the STAT3 signalling cascade since the inhibition of STAT3 with WP1066 impedes the formation of the cachectic condition and the alteration of lactate dehydrogenase and adiponectin levels. Collectively, these findings show that STAT3 is responsible for the altered lactate dehydrogenase and adiponectin levels that, in turn, could participate in the worsening of this pathology and highlight a step forward in the comprehension of the mechanisms underlying the onset of the cachectic condition in adipocytes.

## 1. Introduction

Cancer cachexia is a complex multifactorial syndrome driven by a chronic and systemic inflammatory state characterised by the loss of homeostatic control of energy and protein balance [1,2]. Muscle and adipose tissues are the main target organs during cancer cachexia progression, which is characterised by extensive metabolic changes and inflammation. Adipose tissue is an endocrine organ secreting adipokines that plays a pivotal role in the regulation of whole-body metabolic homeostasis and in several physiological processes [3]. A cachectic condition in adipose tissue provokes some remodelling, including morphological and metabolic changes, ranging from a decrease in cell volume to increased lipolysis and the “browning” process, characterised by a progressive shift from white adipose tissue (WAT) to brown adipose tissue (BAT) [4]. Inflammation underlies these phenotypical and metabolic changes leading to marked adipose tissue wasting and lipid reserve breakdown [3]. Host- or tumour-derived inflammatory factors such as Interleukin-6 (IL-6) play a pivotal role in the progressive switch from WAT to BAT that directly contributes to the increase in energy expenditure [5]. Furthermore, the WAT-to-BAT switch promotes, in turn, lipid mobilization inducing adipose tissue wasting [6]. Although the involvement of adipose tissue in the establishment of cancer cachexia has long been known, the underlying metabolic changes have been little studied so far. Given the prominent role of adipose tissue in the regulation of systemic energy homeostasis, metabolic alterations in adipose tissue could represent a crucial point in both the onset and exacerbation of the cachectic state. Cancer cachexia is associated with altered secretion of hormones, and many cachectic patients manifest a dysregulation of the circulating levels of some adipokines [4]. In particular, adiponectin has key functions in target tissues where adipokines are involved in several processes, such as tissue regeneration [7] and metabolism regulation [8]. Hence, the dysregulation of adiponectin production and secretion by adipose tissue can lead to important consequences for target tissues, where the processes controlled by adiponectin could be greatly altered. 

Previously, we reported that conditioned media from murine colon carcinoma CT26 cells (CM CT26) provoke metabolic alterations that drive the onset of a cachectic condition in myotubes. The hindrance of these metabolic alterations, mainly increased lactate production and decreased oxygen consumption, impedes the onset of cachectic features in myotubes [9]. Moreover, we showed that the addition of sodium pyruvate to CM CT26 impedes the formation of the cachectic state in myotubes, acting on pyruvate dehydrogenase, whose activity is decreased in cachectic muscles and remains unaltered in the presence of sodium pyruvate [10]. 

Here, we report that cachectic adipocytes display increased lactate production and decreased oxygen consumption associated with the increased level of lactate dehydrogenase (LDH) and decreased adiponectin production. These alterations, observed both in murine and human adipocytes, are driven by the STAT3 signalling pathway since the use of the STAT3 inhibitor WP1066 impedes the phenotypic, metabolic, and molecular effects induced by the secretome of colon cancer cells. 

## 2. Results

### 2.1. CM CT26 Treatment Induces Metabolic Alterations and LDH Up-Regulation

The treatment of murine adipocytes with CM CT26 induces a cachectic phenotype, characterised by reduced cell area (Appendix A) and decreased quantities of adipocyte droplets (Appendix A). Moreover, lactate production and oxygen consumption appear altered in CM CT26-treated adipocytes with lower oxygen consumption (Figure 1A) and increased lactate production (Figure 1B). The same metabolic alterations have been observed in human adipocytes (differentiated in vitro from human primary adipose stem cells (ASCs) isolated from visceral adipose tissue biopsies) treated with CM CT26 (Figure 1D,E). These metabolic alterations are associated with LDH up-regulation in both murine (Figure 1C) and human (Figure 1F) adipocytes due to CM CT26 treatment. These findings demonstrate that CM CT26 promotes cachectic features and metabolic perturbations associated with increased LDH content in murine and human adipocytes.

### 2.2. Metabolic Alterations Are Involved in the Formation of Cachectic Features in Adipocytes

To dissect the role of metabolic modifications in the onset of cachectic features, the LDH inhibitor oxamate (50 mM) was added to CM CT26. Figure 2 shows that oxamate impedes the acquisition of both phenotypic and metabolic cachectic features in adipocytes. The adipocytes treated with CM CT26 containing oxamate displayed lipid droplet widths and cell area similar to the control cells (Figure 2A–C). Moreover, oxamate hindered the increase in lactate production (Figure 2D), the upregulation of LDH (Figure 2E), and the decrease in oxygen consumption (Figure 2F) induced by CM CT26. These findings suggest that increased lactate production plays a role in the induction of cachexia in adipocytes, as highlighted by the beneficial effects provided by oxamate.

### 2.3. CMCT26-Treated Adipocytes Show Decreased Adiponectin Levels

We assayed the level of adiponectin, one of the main adipokines secreted by adipose tissue. Murine adipocytes treated with CM CT26 displayed a greatly decreased level of adiponectin synthesis (Figure 3A) and secretion (Figure 3B) since a lower amount of adiponectin was observed in the medium in comparison with that secreted by the control adipocytes. The same effect was observed after the treatment of adipocytes with IL-6, a cytokine greatly involved in the onset of cancer cachexia. Appendix A shows that IL-6 treatment provokes in adipocytes the same consequences induced by CM CT26 regarding myotube phenotype (Appendix A), lactate production (Appendix A), and oxygen consumption (Appendix A). The IL-6-treated adipocytes displayed decreased adiponectin production like the cells treated with CM CT26 (Figure 3C). A similar result was obtained for the human adipocytes treated with CM CT26 (Figure 3D).

Overall, these findings show that CM CT26 or IL-6 treatment greatly affects adiponectin production and secretion in both murine and human adipocytes.

### 2.4. Phenotypic and Metabolic Alterations Involve the STAT3 Signalling Pathway

The STAT3 signalling pathway plays a prominent role in the induction of cancer cachexia [10,11]. CM CT26 induces the activation of STAT3 in adipocytes after five minutes of stimulation, as shown in Figure 4A. To dissect the role of STAT3 in the acquisition of cachectic features, adipocytes were treated with CM CT26 containing the STAT3 inhibitor WP1066. Appendix A shows the effectiveness of WP1066, which impedes the phosphorylation of the Tyr705 of STAT3 due to CM CT26 stimulation. STAT3 inhibition hinders the acquisition of both phenotypic alterations due to CM CT26, thus impeding the decrease in lipid droplet width (Figure 4B,C) and cellular area (Figure 3D). Regarding metabolism, WP1066 impedes the changes in lactate production and oxygen consumption, which remain similar to those for the control adipocytes (Figure 4E,F).

These findings show that the activation of the STAT3 cascade by CM CT26 plays a role in the acquisition of cachectic characteristics and metabolic modifications in adipocytes.

### 2.5. STAT3 Signalling Drives LDH and Adiponectin Levels in CMCT26-Treated Adipocytes

We analysed the involvement of STAT3 in the alterations of LDH and adiponectin levels in the CM CT26-treated adipocytes. The treatment of adipocytes with CM CT26 containing the STAT3 inhibitor WP1066 impeded the up regulation of LDH (Figure 5A) and the downregulation of adiponectin (Figure 5B) provoked by the CT26 secretome, thus showing that the alterations of these two proteins are controlled by STAT3 signalling.

## 3. Discussion

Our paper shows that the secretome from murine colon carcinoma CT26 cells provokes the acquisition of cachectic features and metabolic alterations, such as a decrease in oxygen consumption and increased production of lactate, in adipocytes. These adipocytes display enhanced LDH and decreased adiponectin levels. The STAT3 signalling cascade underlies these modifications since its inhibition precludes the formation of the phenotypic, metabolic, and molecular alterations, according to the model shown in Figure 6. We highlight that STAT3 is responsible for the altered levels of both LDH and adiponectin that, in turn, could participate in the onset and/or worsening of the cachectic condition.

The main morphological changes in cachectic adipocytes, observed in animal models and cachectic patients, are a decrease in cellular perimeter and area [12]. Concurrently, a great deal of lipid breakdown occurs in the cachectic condition, leading to a decrease in lipid droplet width and increased levels of circulating free fatty acids that correlate with the worsening of the pathology [4]. Accordingly, our results demonstrate that the CT26 secretome provokes morphological modifications typical of cachectic adipocytes such as a reduction in lipid droplet width and cell area.

An intriguing result is the LDH up-regulation induced by the CT26 secretome in adipocytes. LDH is an intracellular enzyme that, under hypoxic conditions, catalyses the reversible conversion of pyruvate in lactate with the associated oxidation of NADH to NAD+. LDH is a tetramer composed of a combination of the A and B subunits that generate five isoenzymes in humans [13]. Deregulated LDH levels have been observed in several tumours, suggesting that LDH is a tumour-promoting enzyme. Increased LDH levels have been implicated in different processes in cancer cells, such as the epithelial-to-mesenchymal transition [14,15] and invasion/migration [16,17]. Conversely, LDH inhibition decreases tumour growth and metastasis formation [18,19]. Enhanced LDH levels provoke the secretion of an increased amount of lactate in the extracellular environment that affects the behaviour of several immune cell populations. For example, lactate accumulation inhibits T lymphocyte functions [20], immune surveillance by T and natural killer (NK) cells [21], and innate and adaptive immunity through the recruitment of myeloid-derived suppressor cells [22]. Our results show that the CT26 secretome induces LDH up-regulation and increased lactate secretion, suggesting that LDH could be involved in the acquisition of cachectic features in adipocytes. This hypothesis was confirmed using the LDH inhibitor oxamate. The addition of oxamate to CM CT26 hampers the effects induced by the CT26 secretome since adipocytes are similar to untreated cells in terms of droplet width and adipocyte area, oxygen consumption, and lactate production. Concerning LDH, the treatment with oxamate induced an up-regulation of this enzyme’s levels both in the control and CM CT26-treated adipocytes, probably as a compensatory response of adipocytes to LDH inhibition by oxamate. Accordingly, both the control and CM CT26-treated adipocytes displayed decreased lactate production in comparison with their untreated counterparts.

Adipose tissue is an endocrine organ that produces and secretes a plethora of hormones called adipokines, including adiponectin. Adiponectin is produced as a monomer that can be assembled for the formation of different complexes [23]. Adiponectin circulates at high concentrations in the bloodstream, and it targets several tissues, where it has different functions, ranging from metabolic to differentiative roles [7]. Concerning metabolism, adiponectin enhances glucose uptake, activates glycolysis, and promotes triglyceride breakdown [24], while it acts as differentiating hormone in several tissues, such as skeletal muscle, bone, endothelium, kidney, and stem cells [7]. There are contrasting data about adiponectin levels in cachectic conditions depending on cancer type and cachectic phase [4]. Our results show that CM CT26 induces the downregulation of adiponectin levels in treated adipocytes with a consequent reduction in adiponectin levels in the extracellular environment. This result was observed in both murine and human adipocytes after CM CT26 treatment, thus demonstrating that the effect is general and not restricted to a specific cell type. Based on the importance of adiponectin in the target tissues, we can speculate that the decrease in adiponectin production by cachectic adipocytes affects the functionality of the target organs, which is probably associated with the worsening of the pathology in question.

This study shows that the effects induced by the CT26 secretome are driven by the STAT3 signalling cascade. STAT3 belongs to a family of proteins that transduce signals from activated receptors to the nucleus by activating and regulating the transcription of target genes. The involvement of STAT3 signalling has been demonstrated in several human cancers (where it correlates with a poor prognosis) [25] and in cancer cachexia [26]. In the latter, the STAT3 pathway is involved in the onset of cachexia in both skeletal muscle and adipose tissue [27], where lipolysis and wasting occur through the activation of the STAT3 signalling pathway [28,29]. Moreover, our recent findings showed that activated STAT3 is involved in the acquisition of cachectic features in myotubes. In this model, the cachectic condition brought about by CM CT26 treatment could be impeded by the addition of pyruvate, which blocks the phosphorylation/activation of the STAT3 cascade [10]. STAT3 contains two phosphorylation sites that induce STAT3 phosphorylation/activation: a tyrosine residue on the SH2 domain (Tyr705) and a serine residue on the transactivation domain (Ser727). Several observations have shown that the phosphorylation of Tyr705 in STAT3 is involved in a plethora of diseases. For example, phosphorylated Tyr705 contributes to systemic inflammation and coagulopathy in sepsis, while the pharmacological inhibition of Tyr705 phosphorylation alleviates these symptoms [30]. Moreover, in rat hearts, diabetes decreases STAT3 activation due to Tyr 705 phosphorylation [31], while the differential phosphorylation of STAT3 at Tyr705 and Ser 727 regulates the motile programs in cancer stem cells for the metastatic process [32]. Here, we showed that CM CT26 induces STAT3 phosphorylation and that the STAT3 inhibitor WP1066 impedes secretome effects. CT26 secretome contains many cytokines (as IL-6) that drive the modifications induced by CM CT26 [9]. In adipocytes, the CT26 secretome acts by activating the STAT3 signalling cascade, which is responsible for the phenotypic and metabolic alterations observed in CM CT26-treated adipocytes. Interestingly, our findings demonstrate that STAT3 is responsible for LDH up-regulation and adiponectin down-regulation, as observed in the adipocytes treated with CM CT26 secretome containing WP1066, where both protein levels were like those observed in the control cells. To date, no data report the involvement of STAT3 in the transcriptional control of LDH and adiponectin genes. Particularly, the up regulation of LDH leads to an increased production of lactate that could have a potential role in the onset and exacerbation of the cachectic condition in adipocytes. Although the role of secreted lactate in the cancer environment is well known [33], little is known about the effect of this metabolite in cancer cachexia. Regarding adipose tissue, emerging research is highlighting lactate’s function as a metabolite and signalling molecule playing a fundamental role in adipocytes [34,35], and a possible role in cachexia onset cannot be excluded.

These findings suggested a model that is illustrated in Figure 6. The treatment of adipocytes with CT26 secretome induces the phosphorylation of STAT3 (Tyr705), which participates in adiponectin down-regulation and the LDH up-regulation, responsible for the increased lactate amount in the extracellular environment. The blocking of STAT3 action through the inhibitor WP1066 impedes the downstream effects induced by CM CT26, such as the acquisition of the cachectic phenotype and metabolic modifications.

It is important to highlight that cancer cachexia is driven by the same mechanisms in both myotubes [9] and adipocytes. In both models, we demonstrated that increased lactate production drives the formation of the cachectic condition, suggesting that LDH and lactate are an enzyme and a metabolite involved in the induction of this pathology, thus opening the way for the use of new targets for the creation of novel strategies for counteracting cancer cachexia.

## 4. Materials and Methods

### 4.1. Materials

Murine 3T3L1 preadipocytes and CT26 colon carcinoma cell lines were obtained from ATCC. Unless otherwise specified, all reagents were purchased from Sigma-Aldrich (St. Louis, MO, USA); anti-Adiponectin (ab22554) primary antibody was obtained from Abcam (Cambridge, UK); anti-LDH (sc-33781) and anti-STAT3 (sc-8019) primary antibodies and STAT3 Inhibitor III WP1066 (sc-203282) were purchased from Santa Cruz Biotechnology (Dallas, TX, USA); anti-phospho-Tyr705-STAT3 (#9145) primary antibody was purchased from Cell Signalling Technology Inc. (Danvers, MA, USA); SDS-PAGE materials and ECL reagents were obtained from Bio-Rad Laboratories (Hercules, CA, USA); and K-LATE kit for lactate assay was purchased from Megazyme (Bray, Ireland).

### 4.2. Methods

#### 4.2.1. Cell Culture and Adipocyte Differentiation

Murine 3T3L1 preadipocytes and CT26 murine colon carcinoma were grown in DMEM supplemented with 10% FBS (growing medium) in a 5% CO_2_ humidified atmosphere. For the differentiation of 3T3L1 into adipocytes, 3 × 10^3^ cells per cm^2^ were seeded and grown until reaching about 70% confluence. Cells were maintained for three days in MDI induction medium composed of 0.5 mM of 3-isobutyl-1-methylxanthine, 1 µM of dexamethasone, and 10 µg/mL of insulin. After three days, MDI induction medium was replaced with DMEM containing 10% FBS and 10 µg/mL of insulin. After an additional three days, insulin-containing medium was replaced with DMEM containing 10% FBS that had been replaced every two days until fully differentiated adipocyte-like cells were obtained. Human primary adipose stem cells (ASCs) were isolated from the stromal vascular fraction derived from visceral adipose tissue biopsies conducted after receiving written informed consent (Local Ethical Committee’s approval Ref. Protocol 58-11 03/06/2011), as described elsewhere [36]. ASCs were cultured in DMEM supplemented with 20% FBS, 2 mM of L-glutamine, 100 U/mL of penicillin, 100 µg/mL of streptomycin, and 1 µg/mL of Amphotericin-B. Cells were incubated at 37 °C in a humidified 5% CO_2_ atmosphere. For the experiments, ASCs between passages 2 and 7 were used. For adipose differentiation, 5 × 10^4^ ASCs were plated in 6-well plates and induced to differentiate into mature adipocytes using a differentiation medium (DMEM containing 10% FBS, 0.5 mM 3-isobutyl-1-methylxanthine, 1 µM dexamethasone, 10 µM insulin and 1 μM rosiglitazone) for 15 days [36]. ASCs cultured for the same period in DMEM containing 10% FBS were used as a control.

#### 4.2.2. Preparation of Conditioned Media (CM)

CT26 colon carcinoma cell line was cultured in growing medium until sub-confluence. Cells were then washed twice with PBS, and then serum-free medium was added for 48 h. The CM CT26-conditioned medium (CM CT26) was centrifuged at 1500 rpm to remove debris and immediately used for adipocyte treatment or stocked at −80 °C. For the treatment of both murine and human adipocytes, CM CT26 was diluted in a 1:2 ratio with free-serum DMEM.

#### 4.2.3. Measurement of the Width and Area of Lipid Droplets

Measurements of the width and area of lipid droplets were obtained using ImageJ software version 1.8 At least ten randomly chosen fields for each sample were used, and, for each adipocyte, a minimum of 20 lipid droplets were measured. The mean value is reported in the bar graph.

#### 4.2.4. Lactate Assay

Lactate assay was performed in culture medium using the K-LATE kit according to the manufacturer’s instructions. The amount of lactate (mg/mL), exclusively produced from treated adipocytes, was obtained by subtracting the amount of lactate in CM CT26. Data were normalized using total protein content in each sample, thus obtaining mg/mL of lactate.

#### 4.2.5. Analysis of Oxygen Consumption

Cells were suspended in 1 mL of growing medium and transferred to an airtight chamber maintained at 37 °C. A Clark-type O_2_ electrode (Oxygraph Hansatech, Norfolk, England) was used to assay adipocyte oxygen consumption over 10 min. Oxygen consumption rate (OCR), measured as nmol/mL/min, was obtained, and the OCR was normalized with respect to the total protein content of each sample and is reported in the bar graph as nmol O_2_/min/mg.

#### 4.2.6. Immunoblot Analysis

Immunoblot analysis was performed as already reported [10]. Briefly, samples were lysed in complete radio-immunoprecipitation assay (c-RIPA) buffer (150 mM of NaCl, 100 mM of NaF, 2 mM of EGTA, 50 mM of Tris HCl with a pH of 7.5, 5 mM of orthovanadate, 1% Triton, 0.1% SDS, and 0.1% protease inhibitor cocktail). Lysates were then centrifuged at 14,000 rpm at 4 °C for 10 min, and the supernatant was recovered. Total protein content of each sample was obtained using Bradford assay (Bio-Rad Laboratories, Hercules, CA, USA). The same quantities of proteins for each sample were separated using SDS-PAGE and transferred onto PVDF membranes. PVDF membranes were incubated with a specific primary antibody in 2% non-fat dry milk solution (PBS, 0.05% tween) at 4 °C for 24 h and with horseradish-conjugated secondary antibody for 1 h at room temperature. Specific protein bands were detected using chemiluminescence reactions induced by probing PVDF membranes with ECL (Bio-Rad Laboratories, Hercules, CA, USA) and quantified using ImageJ software (version 1.8). Coomassie-stained PVDF membranes were used for normalization, and the mean values are reported in the bar graph as arbitrary units.

#### 4.2.7. Statistical Analysis

Data are presented as means ± SD from at least three independent experiments. Statistical analysis of the data was performed using Student’s *t* test or via one-way ANOVA conducted using GraphPad Prism (Graphpad Holdings, LLC, San Diego, CA, USA) version 6.0. A *p*-value < 0.05 was considered statistically significant.

## Figures and Tables

**Figure 1 ijms-24-16343-f001:**
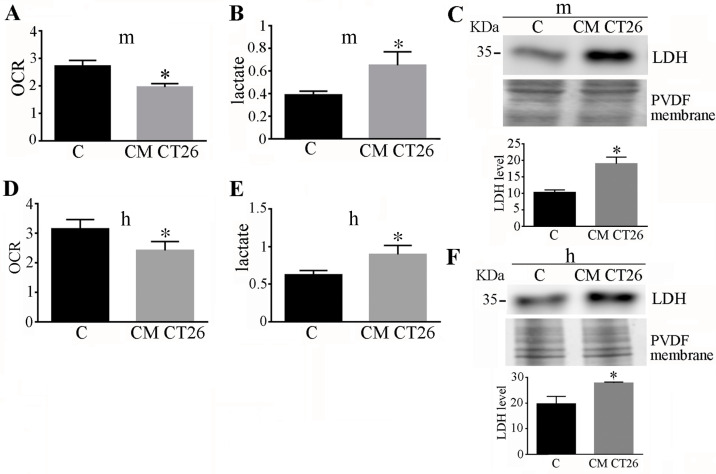
The treatment of murine and human adipocytes with CM CT26 induces metabolic alterations and LDH up-regulation. Adipocytes were treated with CM CT26 for 48 h. (**A**,**D**) Analysis of oxygen consumption reported as oxygen consumption rate (OCR); (**B**,**E**) lactate assay; (**C**,**F**) LDH immunoblot. PVDF membrane was used for normalization, and the mean values are reported in the bar graph. C, control adipocytes; m, murine adipocytes; h, human adipocytes. CM CT26: conditioned media from colon cancer cell CT26; LDH: lactate dehydrogenase; * *p* < 0.05; n = 3.

**Figure 2 ijms-24-16343-f002:**
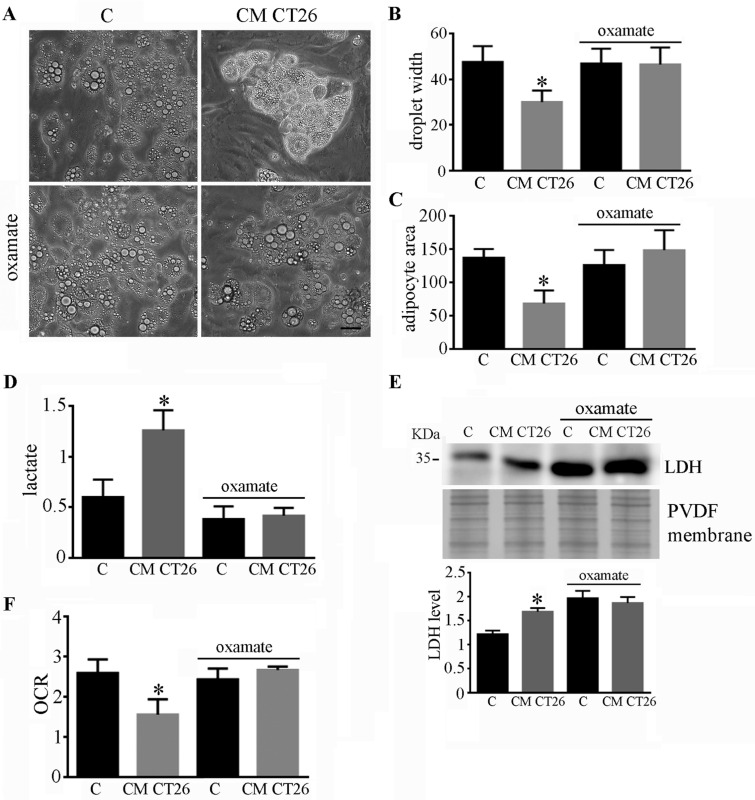
Inhibition of LDH by oxamate hinders the effects induced by CM CT26. Murine adipocytes were treated with CM CT26 for 48 h. Where indicated, oxamate (50 mM) was added to CM CT26 and to the control. (**A**) Representative images of adipocytes. (**B**) Measures of droplet width and (**C**) adipocyte area. The parameters in (**B**,**C**) were obtained using at least ten randomly chosen fields, and the mean values are reported in the bar graphs. (**D**) Lactate assay. (**E**) LDH level obtained using immunoblotting. PVDF membrane was used for normalization, and the mean values are reported in the bar graph. (**F**) Oxygen consumption rate (OCR); CM CT26: conditioned media from colon cancer cell CT26; LDH: lactate dehydrogenase; * *p* < 0.05; n = 3; scale: 50 µm.

**Figure 3 ijms-24-16343-f003:**
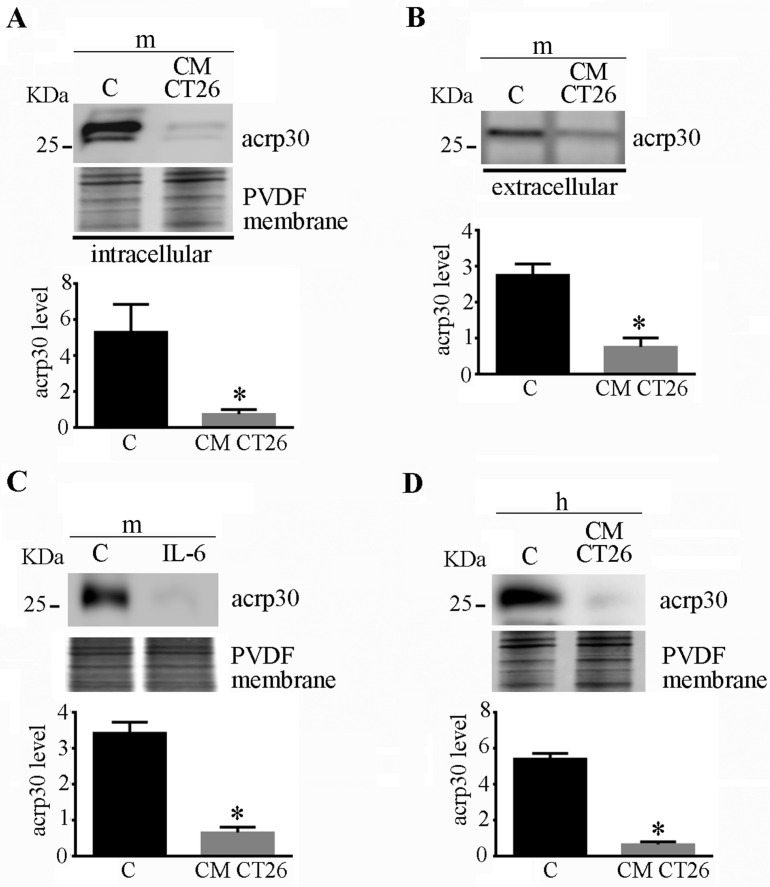
Human and murine adipocytes showing decreased levels of adiponectin. Adipocytes were treated with CM CT26 or with IL-6 (240 ng/mL) for 48 h, and adiponectin levels was detected using immunoblots. (**A**) Intracellular adiponectin level and (**B**) secreted adiponectin in murine adipocytes. (**C**) Intracellular adiponectin in murine adipocytes treated with IL-6. (**D**) Intracellular adiponectin in human adipocytes treated with CM CT26. C, control adipocytes; m, murine adipocytes; h, human adipocytes. CM CT26: conditioned media from colon cancer cell CT26; acrp30: adiponectin; * *p* < 0.05; n = 3.

**Figure 4 ijms-24-16343-f004:**
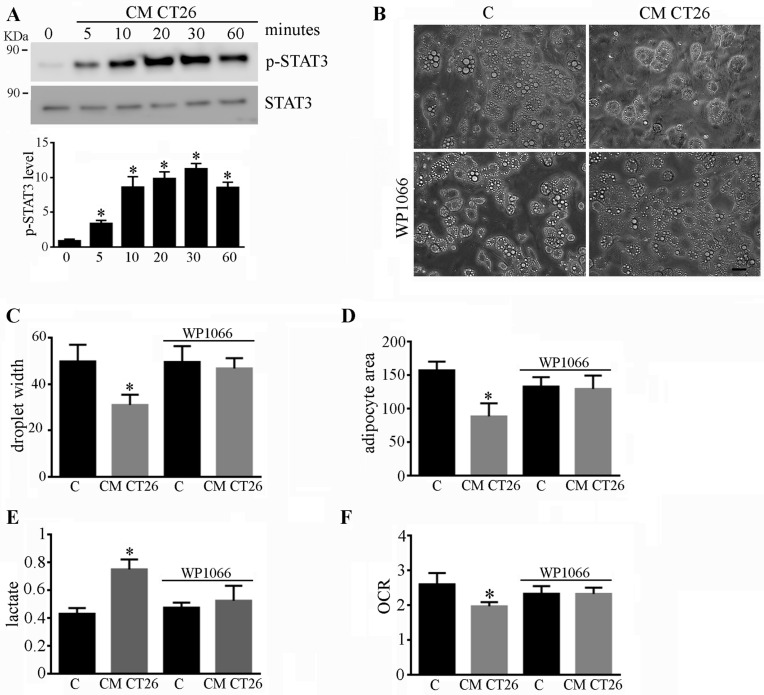
STAT3 signalling drives the effects induced by CM CT26. (**A**) Murine adipocytes were treated with serum-free medium overnight and then stimulated for the indicated time with CM CT26. The levels of total and phosphorylated-Tyr705 of STAT3 were detected using immunoblots. (Panels **B**–**F**) show murine adipocytes that were treated with CM CT26 for 48 h. Where indicated, WP1066 (10 μM) was added to CM CT26 or to the control. (**B**) Representative images of adipocytes. (**C**) Measures of lipid droplet width and (**D**) adipocyte area, both calculated using at least ten fields chosen randomly. (**E**) Lactate assay. (**F**) Analysis of oxygen consumption reported as oxygen consumption rate (OCR). C, control; CM CT26: conditioned media from colon cancer cell CT26; * *p* < 0.05; n = 3; scale: 50 µm.

**Figure 5 ijms-24-16343-f005:**
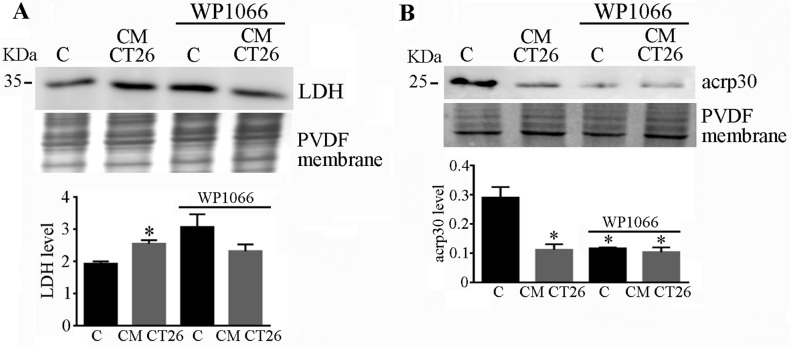
STAT3 controls LDH and adiponectin levels. Murine adipocytes were treated with CM CT26 for 48 h. Where indicated, WP1066 (10 μM) was added to CM CT26. (**A**) LDH and (**B**) adiponectin levels assayed using immunoblotting. PVDF membranes were used for normalization, and the mean values are reported in the bar graph. C, control; CM CT26: conditioned media from colon cancer cell CT26; LDH: lactate dehydrogenase; acrp30: adiponectin; * *p* < 0.05; n = 3.

**Figure 6 ijms-24-16343-f006:**
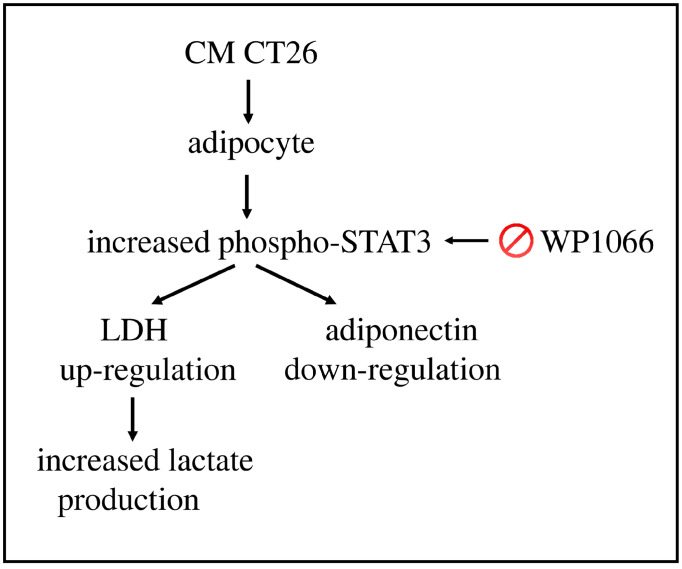
Proposed model of CM CT26 effects in adipocytes. The treatment of adipocytes with CM CT26 induces STAT3 phosphorylation on Tyr 205. The activation of STAT3 cascade induces the increase in lactate production and the decrease in oxygen consumption, associated with the enhanced level of LDH and the concomitant decrease in adiponectin. The blocking of STAT3 cascade using the inhibitor WP1066 hinders the down-stream effects due to CM CT26, which becomes ineffective. CM CT26: conditioned media from colon cancer cell CT26.

## Data Availability

No new data were created or analyzed in this study. Data sharing is not applicable to this article.

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
