# Peer review of "STAT3 Signalling Drives LDH Up-Regulation and Adiponectin Down-Regulation in Cachectic Adipocytes"

_ijms, 2023, doi:10.3390/ijms242216343_

Round 1

Reviewer 1 Report

Comments and Suggestions for Authors

Reviewer comments.

In the present manuscript, Michele Mannelli et al. show that STAT3 signaling drives LDH up-regulation and adiponectin down-regulation in cachectic adipocytes leading to STAT3 is responsible for the altered lactate dehydrogenase and adiponectin levels. Using pharmacological inhibitors of oxamate, they mainly report alterations of enhanced lactate production and decreased oxygen consumption. Altogether, the manuscript contains some interesting findings but derived exclusively from studies performed with pharmacological inhibitors with doubtful specificity, especially in vitro. In addition to a complementary genetic approach, some mechanistic aspects would also need to be clarified to strengthen the manuscript and the authors’ conclusions.

1.     It is not clear if metabolic and molecular effects are affecting both in murine and in human adipocytes or cells that have already increased lactate production and how specifically these cells express LDH. It is important to clarify this aspect to support the manuscript’s conclusions.

2.     Can the authors please comment on the fact that in figure 3 the authors show that there is down regulation of CT26 and IL-6 in the absence of CM CT26. Does this impact the overall conclusion?

1.     Fig 5A. Are the concentrations at which the inhibitor reduces LDH proteins levels cytotoxic? It would be important to show cell viability under the conditions used in this assay, LDH levels upon inhibitor treatment to support its specificity.

3.     Can the results found in figure 4B be explained by the other immune modulatory functions already described for CM CT26  and WP1066? The authors should include this in the discussion.

4.     The authors should explain the significance of increased CT26 expression in Fig. 5B.

5.     A couple of typos should be corrected throughout the manuscript, including some in the abstract.

6.     Some rephrasing is also required, e.g. “using similar but still differential study designs

Author Response

Point to point answers to Reviewer 1

In the present manuscript, Michele Mannelli et al. show that STAT3 signaling drives LDH up-regulation and adiponectin down-regulation in cachectic adipocytes leading to STAT3 is responsible for the altered lactate dehydrogenase and adiponectin levels. Using pharmacological inhibitors of oxamate, they mainly report alterations of enhanced lactate production and decreased oxygen consumption. Altogether, the manuscript contains some interesting findings but derived exclusively from studies performed with pharmacological inhibitors with doubtful specificity, especially in vitro. In addition to a complementary genetic approach, some mechanistic aspects would also need to be clarified to strengthen the manuscript and the authors’ conclusions.

  1. It is not clear if metabolic and molecular effects are affecting both in murine and in human adipocytes or cells that have already increased lactate production and how specifically these cells express LDH. It is important to clarify this aspect to support the manuscript’s conclusions.

We thank the referee for the comment. Both murine and human adipocytes have treated with CM CT26 and metabolic and LDH increase has ben observed in both cells. According to the referee’ suggestion we modified the paragraph 2.1 of the results, as follows:

line 75: “subjected to the same treatment” has been substituted with “treated with CM CT26”.

  1. Can the authors please comment on the fact that in figure 3 the authors show that there is down regulation of CT26 and IL-6 in the absence of CM CT26. Does this impact the overall conclusion?

We thank the referee for the comment. If I understand correctly, the referee is referring to the level of acrp30 in control cells (indicated with C) that are not treated with CM CT26. In these samples no down regulation of the protein has been observed. The acrp30 level is similar in Figures 3A, 3C and 3D (showing the intracellular acrp30 level), while Fig. 3B shows extracellular acrp30, which could have a different level in comparison to intracellular amount. For this, the overall conclusions are not affected.

  1. Fig 5A. Are the concentrations at which the inhibitor reduces LDH proteins levels cytotoxic? It would be important to show cell viability under the conditions used in this assay, LDH levels upon inhibitor treatment to support its specificity.

In this paper, we used two inhibitors: oxamate for LDH and WP1066 for blocking the STAT3 signaling cascade. The viability of the cells treated with a single inhibitor has been evaluated using MTT assay, before carrying out the experiments.  This assay did not highlight any cellular suffering at the doses used. Moreover, for the STAT3 inhibitor WP1066, its efficacy in STAT3 activation is shown in Suppl. Fig. 3.

  1. Can the results found in figure 4B be explained by the other immune modulatory functions already described for CM CT26 and WP1066? The authors should include this in the discussion.

We thank the referee for the comment. Fig. 4B shows representative images of untreated and CM CT26-treated adipocytes with or without the STAT3 inhibitor WP1066. As described in the paper (Paragraph 2.4), CM CT26 induces the cachectic features in adipocytes as the decrease of adipocyte area and lipid droplet width. CM CT26 effects are impeded by WP1066 demonstrating that STAT3 signaling drives these modifications. As suggested by the referee, we add in the Discussion as follows (lines 244-248).

CT26 secretome contains many cytokines (as IL-6) that drive the modifications induced by CM CT26 (9). In adipocytes CT26 secretome acts by activating STAT3 signaling cascade, which is responsible for the phenotypic and metabolic alterations observed in CM CT26-treated adipocytes.

  1. The authors should explain the significance of increased CT26 expression in Fig. 5B.

We thank the referee. I apologize, I do not understand the meaning of the “CT26 expression”. CT26 are the murine colon cancer cells and in the figure is shown the conditioned media obtained culturing these cells. Figure 5B shows that STAT3 signaling controls adiponectin level in adipocytes treated with CM CT26. This result is discussed in the Discussion section (lines 248-252).

  1. A couple of typos should be corrected throughout the manuscript, including some in the abstract.

We thank the referee. The typos have been corrected.

  1. Some rephrasing is also required, e.g. “using similar but still differential study designs

We thank the referee. Some rephrasing has been performed.

Reviewer 2 Report

Comments and Suggestions for Authors

This is an interesting article on cancer cachexia, a multifactorial syndrome characterized by progressive loss of muscle mass in a cancer patient. Cancer growth and anticancer treatments cause harmful adaptations in patients, often resulting in paraneoplastic syndromes, such as neoplastic cachexia. Cachexia is characterized by inflammation and metabolic stress in different organs, resulting in impaired tissue function, reduced tolerance to chemotherapy and poor immune response: all factors that contribute to compromising the quality of life and reducing survival.

In this regard, Mannelli et al. report that in vitro, cachectic adipocytes display a rise in lactate production and a decrease in oxygen consumption due to an increase in lactate dehydrogenase (LDH) levels and a decrease in the production of adiponectin.

In particular, their findings indicate that the secretoma of CT26 cells in mouse colon carcinoma causes adipocytes to acquire cachectic characteristics and metabolic changes, for instance, there is a decrease in oxygen consumption and an increase in lactate production. There is a rise in LDH and a fall in adiponectin in these adipocytes. These changes are the result of the STAT3 signal cascade. The altered levels of both LDH and adiponectin caused by STAT3 could lead to the onset or worsening of the cachectic condition.

The article is generally well written and structured. The introduction provides sufficient background and include all relevant references. The findings are carefully and clearly described. However, I have a few observations:

1.      All abbreviations used in the figures should be explained in the respective legends (if not explained in the figures/tables already).

2.      English revision is needed. The manuscript could benefit from English editing and synthesis (highlight main ideas).

The manuscript will benefit from minor English editing. Please, highlight your key ideas, since they can be missed given the amount of text.

Considering the scientific value and soundness, I recommend to publish the manuscript.

Comments on the Quality of English Language

English revision is needed.

Author Response

Point to point answers to Reviewer 2

This is an interesting article on cancer cachexia, a multifactorial syndrome characterized by progressive loss of muscle mass in a cancer patient. Cancer growth and anticancer treatments cause harmful adaptations in patients, often resulting in paraneoplastic syndromes, such as neoplastic cachexia. Cachexia is characterized by inflammation and metabolic stress in different organs, resulting in impaired tissue function, reduced tolerance to chemotherapy and poor immune response: all factors that contribute to compromising the quality of life and reducing survival.

In this regard, Mannelli et al. report that in vitro, cachectic adipocytes display a rise in lactate production and a decrease in oxygen consumption due to an increase in lactate dehydrogenase (LDH) levels and a decrease in the production of adiponectin.

In particular, their findings indicate that the secretoma of CT26 cells in mouse colon carcinoma causes adipocytes to acquire cachectic characteristics and metabolic changes, for instance, there is a decrease in oxygen consumption and an increase in lactate production. There is a rise in LDH and a fall in adiponectin in these adipocytes. These changes are the result of the STAT3 signal cascade. The altered levels of both LDH and adiponectin caused by STAT3 could lead to the onset or worsening of the cachectic condition.

The article is generally well written and structured. The introduction provides sufficient background and include all relevant references. The findings are carefully and clearly described. However, I have a few observations:

  1. All abbreviations used in the figures should be explained in the respective legends (if not explained in the figures/tables already).

We thank the referee for the suggestion. As suggested, we inserted the abbreviations in the figure legends.

  1. English revision is needed. The manuscript could benefit from English editing and synthesis (highlight main ideas).

We thank the referee for the suggestion. The English has been edited.

The manuscript will benefit from minor English editing. Please, highlight your key ideas, since they can be missed given the amount of text.

Considering the scientific value and soundness, I recommend to publish the manuscript.
